# Investigations on the Morphological, Mechanical, Ablative, Physical, Thermal, and Electrical Properties of EPDM-Based Composites for the Exploration of Enhanced Thermal Insulation Potential

**DOI:** 10.3390/polym14050863

**Published:** 2022-02-22

**Authors:** Nasima Arshad, Ghulam Qasim, Abeer M. Beagan

**Affiliations:** 1Department of Chemistry, Allama Iqbal Open University, Islamabad 44000, Pakistan; qasim80pk1@gmail.com; 2Chemistry Department, College of Science, King Saud University, P.O. Box 2455, Riyadh 11451, Saudi Arabia; abeagan@ksu.edu.sa

**Keywords:** ethylene-propylene-diene monomer—EPDM, vulkasil-C, asbestos, carbon fiber fabric, thermal insultation

## Abstract

The most widely used filler in EPDM-based thermal insulation materials is asbestos which is hazardous to health and environment. The main motivation of this study was to develop improved EPDM-based materials by partially or completely replacing asbestos with other fillers. EPDM-Esprene501A and EPDM-Keltan^®^4869DE were used and the effect of three fillers (vulkasil-C, asbestos, carbon fiber fabric) on mechanical, ablative, physical, thermal, and electrical performances have been investigated. Samples were divided into phase -1, -2, and -3 by compounding EPDM with various percentages of fillers and other necessary ingredients. It was observed that asbestos and carbon fiber in the absence of vulkasil-C did not import enough reinforcement to EPDM-matrix. Experimental evidence showed that presence of vulkasil-C has not only enhanced mechanical properties but also improved thermal and ablation performance of EPDM-based composites. The swelling index was found comparatively lower with vulkasil-C than that with other fillers. Among two EPDMs, EPDM-Esprene based composites have shown comparatively better performance. Among all (phase-1–3) samples, E100K0VA (phase-2) has shown greater mechanical (stress 3.89 MPa; strain 774%), ablative (linear 0.1 mm/s; mass 0.05 g/s), and thermal (material left 91.0%) properties. Overall findings indicated improved properties of EPDM in the presence of vulkasil-C and may help to develop better heat resistant materials.

## 1. Introduction

Thermal protective systems utilize different chemicals which must withstand very high temperatures and pressure. These systems have broad usage in automotive weather-stripping and seals, radiators, belts, boilers, electrical insulation, roofing membranes, rubber mechanical goods and plastic impact modification. The utility of thermal protective materials in various industries is to protect the internal structure of a system from extreme heating by inhibiting the conduction of heat inside the system. For example, central heating systems require long pipes in which hot water circulates. Protecting these pipes with thermal protection sheets will save lot of resources. Also, in geysers, thermal insulating sheets could help to avoid heat loss.

The most important uses of this system are in aerospace vehicles and the defense industries, and its increasing demand has inspired scientists and engineers to develop new thermal protection systems that can resist higher temperatures and pressures [1,2,3,4]. During the rocket propulsion, the insulation protects the SRM case from the heat generated by the propellant combustion and it is typically formulated in such a way that it can bear a high temperature of about 2760 °C and interior pressure of 1500 psi [5].

One of the most important parameters of insulation composites is that they must possess a reasonable shelf life so that they remain intact until used in an insulation system. It is an essential requirement as the manufactured insulating materials may have to wait in storage for long period before its use. As the insulation material faces extreme temperature and pressure it is crucial to check the formulation of the material to prevent any catastrophic failure [6]. Insulation consists of matrix and filler, and both differ in composition and structural integrity. Numerous fillers can be included in the rubber matrix and each of these materials has its own physical and chemical properties affecting the polymer system features added to it [7,8].

The ethylene-propylene-diene monomer (EPDM) has a significant influence on thermal and ablation performances and its structural features provide excellent weather and chemical resistance [9,10]. Oxidative cleavage of double bonds reduces polymer molecular mass and results in loss of desirable physical properties. High molecular weight EPDM is used primarily in the construction and automotive markets and intermediate molecular weight EPDM products provide advantages in molded and extruded applications, whereas low molecular weight products ranging from liquid oligomers to polymers find use as reactive plasticizers, encapsulants, viscosity modifiers, synthetic oil components, adhesives, and sealants [11,12]. Moreover, phenolic antioxidants stabilize the EPDM to ensure its storage [13]. When stored in a cool, dark environment, EPDM has a long shelf life.

In polymer systems, fillers not only reduce the cost of the material but also reinforce the rubber and improve its properties. Among various fillers, asbestos is widely used as heat-resistant material. However, asbestos is carcinogenic and has serious environmental problems [14]. The improved mechanical, and morphological properties of EPDM blends with some other fillers including nano clay [15], silicon carbide (SiC), carbon fiber (CF) [16], and nano-silica [17] have also been reported in different studies. In present research we tried to develop low-cost, low-density EPDM-based materials by using carbon fiber fabric, asbestos and vulkasil-C fillers and studied these materials for improved thermal insulation performance. The physical forms of EPDMs and fillers used in this study are shown in Figure 1.

## 2. Experimental Section

### 2.1. Materials

The materials used in this study were EPDM-Esprene 501 A, EPDM-Keltan^®^ 4869C DE, sulfur, diphenyl guanidine (DPG), 2-mercaptobenzothiazole (MBT), tetramethylthiuram disulfide (TMTD), zinc oxide (ZnO), stearic acid, N-phenyl-2-naphthylamine (PBN), phenyl-β-naphthylamine, dioctyl phthalate (DOP), carbon fiber fabric (CFF, graphite fiber), asbestos and Vulkasil^®^-C (precipitated silica). The EPDM rubbers were purchased from Asia Pacific (SK Global Chemical Co. Ltd., Seoul, South Korea), while other chemicals were purchased from Sigma−Aldrich (Darmstadt, Germany).

### 2.2. Preparation of EPDM-Based Composites

Adequate mixing of EPDM rubber and all ingredients was done by a two-roll mill machine (X(S)K-560, Huahan Machine, Zhuanghe, China). For phase-1 composites, addition of ingredients into EPDM rubber, kneading and mixing was done according to ASTM D3187 [18]. The EPDM-Esprene was passed between the two rolls several times decreasing the nip between the two rolls to the extent of 0.5–1 mm. To the masticated EPDM- Esprene, EPDM-Keltan was added. The blended EPDMs were mixed till homogenization and sulfur was added as a vulcanizing agent followed by mixing for homogenization of the materials at room temperature. Then stearic acid and zinc oxide were added as vulcanization activators. Half Vulkasil-C filler with half plasticizer DOP were added across the mill at a uniform rate and when this portion of the filler has been completely incorporated, the distance between the rolls was opened to 1.65 mm. The remaining Vulkasil-C filler and plasticizer DOP were evenly added across the rolls at a uniform rate. Finally, curing accelerator mercaptobenzothiazole (MBT), DPG, and antioxidant phenyl-β-napthyl amin (PBN) were added. The rolls were set at 0.8 mm followed by mixing for homogenization of the materials. The same procedure was adopted with asbestos and carbon fiber fabric fillers. Similar mixing procedure was adopted for phase-2 and phase-3 composites. However, in phase-2 composites, EPDM-Keltan and EPDM-Esprene were used individually in E_0_K_100_VA and E_100_K_0_VA samples, respectively, along with the fixed quantities of fillers vulkasil-C (30 g) and asbestos (90 g). Similarly, in phase-3 composites, the mixing process for the samples EV_0_A_0_ and KV_0_A_0_ was done without fillers. Vulcanization of each sample was carried out at temperature of 160 °C with optimum curing time. The rubber sheet, which was prepared after mixing, was put into the die of a press machine with the help of the Jannif tool and 10 MPa pressure was applied for 50 min. The schematic representation for the sequential addition of ingredients during the preparation of EPDM-based composites is provided as Figure 1 and the comprehensive detail of different recipes is provided in Table 1.

### 2.3. Morphological Analysis

The morphological studies were carried out with the help of a JSM-6490 scanning electron microscope (SEM, JEOL, Peabody, MA, USA) and chemical characterization was done by energy-dispersive spectroscopy (EDS). The surfaces were made conductive by coating with gold and then examined under the SEM. The images were obtained at a tilt angle of 0° with an operating voltage of 20 kV at 0 °C [19].

### 2.4. Mechanical Analysis

After curing, two sheets of each rubber were prepared for mechanical testing (stress and strain) with the dimension (L × W × T) as 150 mm × 110 mm × 2 mm. From these sheets, dumbbell-shaped samples were cut for mechanical analysis, Figure 2. The mechanical properties like stress and strain were studied with the help of a Universal Tensile Testing Machine (UTM, Instron 5946, Washington, DC, USA) according to DIN 53504/ASTM D412 standard at 20 °C [20]. Each sample was tested 3–5 times, and its average was taken. The test speed was kept at 100 mm/min.

### 2.5. Ablation Analysis

The temperature of the press machine was maintained at 135 °C with an empty die. The rubber sheet, which was prepared after mixing, was put into the die and 10 MPa pressure was applied for 6 h. After curing, three test pieces of each rubber were prepared for ablative testing with a thickness of 10 mm and diameter of 30 mm, Figure 2. The oxyacetylene ablation rate was studied with the help of an oxyacetylene ablation rate tester according to the ASTM E285-86 at 3000 °C [21]. During processing, a stream of hot combustion gases was directed along the normal to the specimen until burn-through was achieved. The linear and mass ablation rates of the test samples were determined by using following formulas:(1)$$\mathrm{Linear}\;\mathrm{ablation}\;\mathrm{rate}\;\left(\mathrm{mm}/S\right)=\frac{\mathrm{Initial}\;\mathrm{thickness}\;\left(\mathrm{mm}\right)-\mathrm{Final}\;\mathrm{thickness}\;\left(\mathrm{mm}\right)}{\mathrm{Ablation}\;\mathrm{time}\;\left(S\right)}$$
(2)$$\mathrm{Mass}\;\mathrm{ablation}\;\mathrm{rate}\;\left(g/S\right)=\frac{\left[\mathrm{Initial}\;\mathrm{mass}\;\left(g\right)-\mathrm{Final}\;\mathrm{mass}\;\left(g\right)\right]}{\mathrm{Ablation}\;\mathrm{time}\;\left(S\right)}$$

### 2.6. Analysis for Physical Properties

#### 2.6.1. Density

Density measurement was carried out by the oil immersion method according to ASTM 792 using paraffin oil [22]. The weight of the sample was measured in air and then in oil and the weight difference was calculated. The density was calculated according to the formula:(3)$$\mathrm{Density}\;\left(\frac{g}{{\mathrm{cm}}^{3}}\right)=\frac{\mathrm{Sample}\;\mathrm{weight}\;\left(g\right)}{\mathrm{Weight}\;\mathrm{differance}\;\left(g\right)}\times \;\mathrm{Density}\;\mathrm{of}\;\mathrm{parraffin}\;\mathrm{oil}\;\left(g/{\mathrm{cm}}^{3}\right)$$

#### 2.6.2. Swelling Index

The swelling behavior of the prepared samples was determined by monitoring the changes in mass with time. From each test sample, known weight (W_2_) was immersed in 10 mL toluene in a diffusion test bottle and kept at room temperature for five days. After five days each sample was removed from the bottle and the wet surface was quickly wiped using tissue paper and re-weighted (W_1_). Each test sample was further dried in an oven at 60 °C for 24 h, cooled in a desiccator, and immediately weighed (W_3_). The swelling parameter was then estimated by using the following equation:(4)$$\mathrm{SI}\;\left(\%\right)=\frac{W_{1}-{\;W}_{2\;}}{W_{3}}\times 100$$
where W_1_, W_2_, W_3_ reflects swollen weight, initial weight, and de-swollen weight, respectively. The experimental setup for swelling index measurement of all the test samples are provided in Appendix A.

### 2.7. Thermal Analysis

The thermogravimetric analyzer (TGA Q50, TA Instruments, New Castle, DE, USA) was used to analyze the weight changes with heat to determine the degradation temperature of the composite samples. Between 3–7 mg of each sample was taken and TGA analysis experiments were carried out from 25 °C to 600 °C. The heating rate was kept 10 °C/min and nitrogen flow rate of 60 mL/min was kept constant in all the cases.

### 2.8. Analysis for Electrical Properties

For the precise voltage and current findings, a 2420-Source Meter^®^ (SMU) instrument (Keithley, Emin, Myanmar) was used to determine the electrical properties of each solid test sample. The current- potential measurements were then used in Ohm’s formula (R = V/I), and conductance was determined as reciprocal of the electrical resistance (K = 1/R). The cell constant is a factor that is used to convert the measured conductance to conductivity (Λ = G × K), where Λ is conductivity (S/cm), G is cell constant (cm^−1^) and K is conductance (S).

## 3. Results and Discussion

### 3.1. EPDM-Based Composites

EPDM-based materials were prepared as phase-1, phase-2, and phase-3 composites. In the 1st phase, blending of two EPDMs with three fillers was done while keeping their quantities constant in each sample. The three samples each were prepared with vulkasil-C, with asbestos, and with carbon fiber fabric filler, so the total nine recipes (EPV1–V3, EPA1-3, EPC1-3) were obtained. In each sample, the quantity of filler was in ascending order. In the 2nd phase, seven samples were prepared as E0K100VA, E15K85VA, E30K70VA, E60K40VA, E75K25VA, E85K15VA, E100K0VA. A blend of EPDM with two fillers Vulkasil and asbestos was made by keeping the number of fillers constant while keeping the quantity of EPDM-Esprene in ascending order and EPDM-Keltan in descending order from sample 1 to sample 7. In the 3rd phase, five samples (EV0A0, EV20A50, EV15A45, KV0A0, KV20A50) were prepared, and among them, three samples have EPDM-Esprene and the remaining two samples contained EPDM- Keltan. Varying quantities of two fillers, Vulkasil-C, and asbestos were used in this investigation. In all the prepared samples, the other ingredients such as curing agents, accelerators, activators, and process aids were kept constant.

### 3.2. Morphological Studies by SEM and EDS

To explore the uniform dispersion of ingredients in the polymer matrix, SEM images and EDS spectra of phase -1, -2 and -3 composites have been taken and provided in Figure 3, Figure 4 and Figure 5, respectively. It has been found that as the dose of filler increased, the intensity of the filler also increased as shown in EDS spectra, Figure 3. SEM images showed that the filler was distributed very well within the polymer matrix. The silicon peak in EDS spectra indicated the presence of Vulkasil-C. The peak height of silicon increased from EPV1-EPV3, as clearly observed from EDS spectra, which further confirmed the presence of greater quantity of Vulkasil-C in the last EPV sample. In phase-2 composites, EDS spectra confirmed that quantity of Mg, S, and Si remained constant in all the seven samples, Figure 4 (only shown for the three samples, others provided in Appendix A). While, in phase-3 composites, EPDM-Esprene and EPDM-Keltan without fillers have shown no silicon, magnesium, and oxygen, Figure 5. Surface morphology showed that the Esprene-EPDM and Keltan-EPDM have smooth surfaces without fillers. The EDS spectra and SEM images for other samples with fillers are provided in the Appendix A. The surface morphology of all the composites indicated homogeneous distribution of the fillers in the EPDM matrix.

### 3.3. Mechanical Properties

Mechanical properties (tensile strength and tensile strain) of EPDM are considerably affected by the type, size, percentage of filler, and their interactions with the polymer matrix [23,24,25]. The plots of average values of tensile stress and tensile strain of phase -1, -2 and -3 composites are shown in Figure 6, while the data for their values is provided in the Appendix A.

Results of phase-1 composites are shown in Figure 6A. The stress (1.87, 2.20, 5.60 MPa) and strain (445, 454, 855%) values of all the three EPV samples, Appendix A, indicated that the filler Vulkasil-C has maximum stress and strain when 37.5% (EPV-3) of its quantity was used, while EPV-1 and EPV-2, where 12.5% and 25% vulkasil-C was added, respectively, have shown comparatively low values. These results indicated that Vulkasil-C, as a reinforcing filler, may have enhanced the inner strength of the composite that resisted the applied force, which may lead to improve the mechanical properties. Some parts of the EPDM may also be adsorbed physically or may be chemically coupled to Vulkasil-C due to which immobilize EPDM covered the surface of Vulkasil-C leading to a clear increase in the mechanical properties. As shown in Figure 6A and the data provided in Appendix A, that by increasing the quantity of filler asbestos (EPA-1 to EPA-3), the stress increased (1.28–2.73 MPa)) but the strain decreased (208–128%). Results indicated that elongation (strain) of the samples decreased with increased asbestos loading. As elongation is inversely proportional to the tensile strength, the elasticity of the rubber chain may be affected by increasing the quantity of asbestos, hence resulted in more rigidity and low elongations in the samples. In the case of carbon fiber fabric filler (EPC-1, EPC-2, and EPC-3), the trend in mechanical properties is anomalous. In EPC-1 and EPC-2, increasing the quantity of filler increased the stress and decreased the strain. However, further increase in filler showed that both stress and strain increased and for EPC-3 these values were found to be 1.44 MPa and 262% respectively. It could be inferred that increased quantity of carbon fiber fabric may have improved its compatibility with EPDM. In general, the best result was obtained when the quantity of Vulkasil-C was 37.5%, the carbon fiber fabric reinforcement showed inconsistency in the mechanical properties, while with asbestos, composites attained intermediate results.

The effect of loading of Vulkasil-C and asbestos on tensile stress and strain of EPDM composites (phase-2) is shown in Figure 6B. EPDM-Keltan sample (E0K100VA) with two fillers have shown comparatively low mechanical properties than that of EPDM-Esprene sample (E100K0VA) with the same quantity of fillers. I could be diagnosed that EPDM-Esprene may have better compatibility with the fillers as compared to EPDM Keltan. For samples E15K85VA, E30K70VA, E60K40VA, and E85K15VA, increase in the stress and decrease in the strain were observed, while an anomalous behavior was observed for the sample E75K25VA where stress and strain both increased. The decrease in the strain was related to the quantity of a filler in phase-1 composites, however, in phase-2 composites, two fillers were used which combinedly may have affected the elasticity of EPDM. The lowest stress and strain for E0K100VA (2.63 MPa, 338%) and E15K85VA (2.56 MPa, 314%), while their highest values for E100K0VA (3.89 MPa, 779%) could be attributed to better mechanical performance of EPDM-Esprene-based composite.

Furthermore, among EV0A0 (1.13 MPa, 161%) and KV0A0 (0.50 MPa, 104%) of phase-3 composites without fillers, the values have shown better mechanical performance of EPDM-Esprene, Figure 6C. While the addition of fillers in both EPDMs enhanced the mechanical properties, however, among both EPDMs, EPDM-Esprene has shown comparatively better mechanical properties alone and with fillers. In comparison to the reported EPDM-silica nano/microcomposites [24], the stress and strain values of most of the tested composites are comparable and even higher for EPV-3, E100K0VA, E0K100VA, E15K85VA, E30K70VA, E75K25VA, EV20A50, EV15A45, and KV20A50.

### 3.4. Ablative Properties

An ablation test describes the properties of materials in response to heat and flame under controlled laboratory conditions. The samples after ablation test are shown in Figure 2, and plots of linear and mass ablation values are given in Figure 7, while the data is provided in Appendix A. Two types of ablative properties including linear ablation and mass ablation were studied. In linear ablation thickness of the sample was measured before and after the ablation test and in mass ablation mass of the sample before and after ablation was recorded.

The effects of separate loading of Vulkasil-C, asbestos, and carbon fiber fabric (phase-1) on ablative properties of EPDM composites are shown in Figure 7A. Less ablation rate either linear or mass is the main requirements of ablative elastomeric materials [3]. The data indicated that with asbestos the composites (EPA-1–3) have shown the low linear and mass ablation rates. However, by increasing the quantity of asbestos the linear ablation rate increased to some extent, but mass ablation remained the same. The carbon fiber fabric (EPC-1–3) composites showed, to some extent, comparable linear and mass ablation as observed for asbestos, while comparatively greater values of linear and mass ablation rates were observed when Vulkasil-C was used. However, among the three composites of Vulkasil-C (EPV-1–3), a decreasing trend was observed with increasing quantity of Vulkasil-C. It can be inferred that Vulkasil-C may enhance the ablative property of EPDM composite if used in large quantity than that of other fillers.

Linear and mass ablation of vulkasil-C and asbestos loaded EPDM phase-2 composites are shown in Figure 7B. The values of mass ablation rate remained almost the same in both types of EPDMs. However, by increasing the quantity of EPDM-Esprene from 15–75%, an insignificant rise in the linear ablation rate was observed. However, linear ablation rate decreased when EPDM-Esprene was 85–100%.

In phase-3 composites, EPDM-Esprene and EPDM-Keltan have the highest values of linear and mass ablation rate when no filler was used as shown in Figure 7C. Among both, EPDM-Esprene showed comparatively lower values of linear and mass ablation rate than that of EPDM-Keltan. The addition of fillers in either EPDM has shown no significant change and the values of both ablation rates remained the same. However, both ablation rates were decreased in the presence of fillers (Vulkasil-C + asbestos).

According to ASTM, the curing parameters of all the ablative properties discussed above were 135 °C at 10 MPa for 6 h. An attempt was made to change the curing parameters of some samples to see what happens to linear and mass ablation rates when curing was done at 160 °C at 10 MPa for 50 min. The results, as given in Table 2, showed that there was no significant difference in the ablative rates measured at both curing parameters. It can be inferred from the data that we can save time, electricity, and other resources by adopting new curing conditions.

### 3.5. Physical Properties

#### 3.5.1. Density

The graphical data for density is provided in Figure 8. An increase in density was markedly observed in phase-1 samples EPV-3, EPA-3, and EPC-3, having a large quantity of filler. However, asbestos containing (EPA-1–3) samples have comparatively greater densities than that of Vulkasil-C and carbon fiber fabric containing composites (EPV-1–3, EPC-1–3) under the same conditions. In phase-2 composites, the sample containing 100% EPDM-Keltan (E0K100VA) has shown highest, while E85K15VA and E75K55VA have shown lowest densities amongst other phase-2 samples. While, in phase-3 samples, EPDM-Esprene and EPDM-Keltan with no fillers have shown lowest densities, EPDM-Esprene and EPDM-Keltan with 13.3% Vulkasil-C and 33.3% asbestos have almost the same and higher densities, while EPDM-Esprene with 10% Vulkasil-C and 30% asbestos has a relative low density. The graphical data inferred that increase in the density may be due to small size, large surface area and good compaction of the filler with the matrix [26].

#### 3.5.2. Swelling Index (SI)

Swelling is an important physical property that must be considered when developing an ablative elastomeric material. Typical ingredients like water and oil are responsible for swelling which are involved in the manufacturing process of elastomers. However, swelling varies from an almost negligible to a large quantity up to 100–150% depending upon fluid condition and type of elastomer [27]. Mechanical properties of elastomers are highly affected by swelling so over the years extensive research has been carried out to lower or eliminate the degree of swelling to maintain the desired mechanical properties [28,29].

The swelling behavior of phase -1, -2 and -3 samples are shown in Figure 9, while the data is provided in Appendix A. As observed from Figure 9A, all the samples of phase-1 have shown decreasing trend in swelling as the quantity of filler was increased. This can be attributed to the reinforcement of filler into the matrix. Vulkasil-C provided less swelling as compared to asbestos and carbon fiber fabric that may be due to comparatively more compact binding between the matrix and Vulkasil-C, hence absorb less moisture. In phase-2 samples, 0–75% EPDM-Esprene-based composites have shown comparatively low swelling than that of other composites, Figure 9B. Further, in phase-3 samples, EV0A0 and KV0A0 having no fillers showed the highest swelling as compared to other counterparts, Figure 9C. Overall, KV20A50, E0K100VA, and E15K85VA composites have shown minimum swelling index with the values of 63, 79, and 98%, respectively.

### 3.6. Thermogravimetric Studies (TGA)

Thermogravimetric analysis is a technique that analyzes the weight changes because of heat to determine the degradation temperature of the polymer composite. The increasing quantity of filler increases the thermal stability [30,31,32]. Hence, the effect of filler loading on the thermal stability of a matrix was characterized by TGA. The thermal behavior of phase -1, -2 and -3 samples are shown in Figure 10. Phase-1 composites, Figure 10A have shown that all composites were almost stable till 450 °C, and after this temperature a sudden degradation accrued at 500 °C. The quantity of samples left, Appendix A, indicated better resistance to temperature for EPA-3 (29.6%), EPC-3 (29.3%) and EPV-3 (26.4%). In phase-2 composites, the variation in thermal behavior can be seen between 200–500 °C. However, composites containing EPDM-Esprene have shown comparatively enhanced thermal stability than that having EPDM-Keltan. The left quantity was also increased with increased EPDM-Esprene in the composites and found to be greater (91.0%) for E100K0VA. While in phase-3 composites, EPDM-Esprene as well as EPDM-Keltan have shown poor thermal properties without fillers and both EPDMs almost decomposed at about 500 °C. However, thermal properties of phase-3 composites enhanced by the addition of two fillers (Vulkasil-C + asbestos) with the left quantities of 32.4, 29.8, 31.9% for EV20A50, EV15A45, and KV20A50, respectively. In general, EPDM-based phase-2 composites have shown maximum resistance against heat with greater left quantities (61.2–91.0%) than that of phase-1 and -3 composites [24]. Among all the tested composites, E100K0VA of phase-2 composites containing 100% EPDM-Esprene has shown highest thermal resistance. The quantity of material left (%) in TGA analysis for all the samples in three phases (1–3) at 500 °C is provided in Appendix A.

### 3.7. Electrical Properties

Conductivity measurements indicated that only few samples of phase-1 composites have shown conductivity and values for these samples are tabulated in Table 3, while other composites were found electrically insulators as conducted no electricity. Among the samples showing conductivity, comparatively greater conductivity value was obtained for the sample EPC-3. In EPC-3, presence of 90% carbon fiber fabric could make the molecular structure like graphite, hence each carbon atom may be bound covalently to three other carbon atoms hence leaving one delocalized electron per carbon atom. Thus, carbon fiber fabric does conduct electricity as it contains delocalized electrons that carry a charge and are free to move throughout the carbon fiber fabric lattice.

### 3.8. Comparative Analysis

Among EPDM-based compositions with all the fillers, the promising results were obtained for EPDM-Esprene based composites with Vulkacil-C filler—a precipitated silica. Here, in Table 4, a comparison of main parameters is provided for the most promising candidate E100K0VA where Vulkacil-C was partially substituted by asbestos. This comparison is made only with those reported insulating materials that were comprised of EPDM matrix with silica as filler [33,34,35,36,37]. Literature comparison further authenticated our findings, especially for E100K0VA with enhanced/or comparable mechanical, ablative, and thermal properties.

## 4. Conclusions

EPDM-Esprene 501A and EPDM-Keltan^®^ 4869C DE were selected to examine the effect of three fillers, namely Vulkasil-C, asbestos, and carbon fiber fabric. The samples were divided into phase -1, -2 and -3 composites. SEM and EDS analysis confirmed the presence of fillers and other ingredients. Among the two EPDM variants, EPDM-Esprene has shown better compatibility with the fillers as compared to EPDM-Keltan. Vulkasil-C showed comparatively greater reinforcement on EPDM matrix than that of other fillers. Comparatively greater mechanical, ablative, thermal and electrical insulation properties with low swelling index was found in most of the phase-2 composites. EPDM-Esprene- based composites have shown better performance as compared to EPDM-Keltan-based composites. Among all composites, the overall properties of E100K0VA were found to be excellent. Hence, the current findings indicate that these cheap, and less hazardous new composites could be explored as potential candidates for the development of improved EPDM-based thermal insulation materials.

## Data Availability

Not applicable.

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
