# Peer review of "Investigations on the Morphological, Mechanical, Ablative, Physical, Thermal, and Electrical Properties of EPDM-Based Composites for the Exploration of Enhanced Thermal Insulation Potential"

_polymers, 2022, doi:10.3390/polym14050863_

Round 1

Reviewer 1 Report

This paper experimented on EPDM-based composites on morphological, mechanical, ablative, physical, thermal, and electrical properties for enhanced thermal insulation application. This topic is interesting. However, it needs some room of improvement to enhance the quality of the paper. Therefore, I recommend the minor revision before it can be considered to be published.

Several aspects need to be improved:

  1. For abstract, please add and summarized all the main findings with a comprehensive argument which indicate the significant of the results. In addition, please indicate the purpose of the research paper. Add along a compact conclusive statement at the end of the abstract to show the potential of the composites for its applications.
  2. The introduction section is discussing on the backgrounds with current literature surveys. It should be noted that normally 'Introduction' should give comprehensive background through literature survey for the previous study by citing previous published work where-by scientific gaps that exist should be brought out. In paragraph 2, please explain on the literature of composite material by referring to these researches: and 10.3390/coatings11111355 and 10.3390/polym13111701.
  3. In Section 1 also, it has been the amount of references is less in number and have to be added. It is suggested to detail out the background of the recent studies on general polymer composites in general properties as such thermal, mechanical and physical properties by referring and citing these relevant articles such: 1016/j.jmrt.2021.12.122, and 10.3390/nano11092186.
  4. Please add a summarization table and a figure indicate previous findings on comparing EPDM-based composites with other polymer composites with its applications.
  5. For section 2, it is a good writing on methodology section. However, please add on a figure regarding the overall flow chart on the methodology part from procurement of materials up until characterization.
  6. However, for section 2.3, please enlarge the size of Figure 2 and labelled the samples accordingly to observe the different. Also provide a schematic diagram to indicate the size of tensile test. This section also suggested to change the name to Tensile Analysis.
  7. For section 3, discussion should relate the experimental findings with other similar previous literatures. Please correlate them in order to establish a comprehensive discussion and analysis. Due to this issue, please revise whole section 4. Please refer this section with the recent research works on EDS and SEM in morphological analysis conducted as such 10.1016/j.compstruct.2021.114644 and 10.1007/s12221-021-0224-6. Please do make a summarization table to indicate the all outcomes from the experimental work conducted including tensile, abrasive, thermal, density, swelling index, electrical and TGA.
  8. For each result findings, please discuss the outcomes with other literatures related to EPDM-based composites to support the statements. Please revised all subtopic 3 thoroughly.
  9. Moreover, for section 4, each subsection should be linked together to indicate the correlation between each results. Please make a summarization paragraph at the end of each paragraph.
  10. For conclusion part, it is do not reflect what had been achieved including many speculations. It is too long and should be in one paragraph. Hence these need to be suitably modified. It may be remembered that this Section forms a summary of all the major observations/ results obtained. Accordingly, here presentation should consist of the main Results or the observations of the study briefly. Moreover, the authors have to include and highlight also the objectives and novelty of the work. Please indicate the future work may be conducted in this section. Hence better to rewrite this Section based on the comments given in the whole text.
  11. Throughout this paper, there is need for better language throughout the manuscript.
  12. Generally, the paper though contains some interesting results and novelty work, it lacks in its proper presentation in the whole manuscript. In view of these, the paper is highly recommended and should be accepted for publication in the revised form. It is suggested that the authors should revise the paper in the light of above comments/suggestions.

Author Response

Please find attached herewith the replies to reviewer#1 comments in pdf 

Reviewer 2 Report

The comments are in the manuscript. The paper is interesting and well conceived, but the results are not sufficiently supported by the literature, there is a lack of discussion.

Author Response

Please find attached herewith the replies to reviewer#2 comments as pdf

Round 2

Reviewer 2 Report

The paper can be accpeted in this form.

Author Response

Dear Reviewers

I am thankful for accepting our manuscript.

Best regards
